# A Current Update on the Distribution, Morphological Features, and Genetic Identity of the Southeast Asian Mahseers, *Tor* Species

**DOI:** 10.3390/biology10040286

**Published:** 2021-04-01

**Authors:** Faizul Jaafar, Uthairat Na-Nakorn, Prapansak Srisapoome, Thumronk Amornsakun, Thuy-Yen Duong, Maria Mojena Gonzales-Plasus, Duc-Huy Hoang, Ishwar S. Parhar

**Affiliations:** 1Brain Research Institute Monash Sunway (BRIMS), Jeffrey Cheah School of Medicine and Health Science, Monash University Malaysia, Bandar Sunway 47500, Malaysia; Faizul.Jaafar@monash.edu; 2Department of Aquaculture, Faculty of Fisheries, Kasetsart University, Chatuchak, Bangkok 10900, Thailand; ffisurn@ku.ac.th (U.N.-N.); ffispssp@ku.ac.th (P.S.); 3Department of Technology and Industries, Prince of Songkla University Pattani Campus, Pattani 94000, Thailand; thumronk.a@gmail.com; 4College of Aquaculture and Fisheries, Can Tho University, Can Tho City 94000, Vietnam; thuyyen@ctu.edu.vn; 5College of Fisheries and Aquatic Science, Puerto Princesa Campus, Western Philippines University, Puerto Princesa City 5300, Philippines; mojenagonzales@yahoo.com; 6Department of Ecology and Evolutionary, Faculty of Biology and Biotechnology, University of Science, Ho Chi Minh City 700000, Vietnam; hdhuy@hcmus.edu.vn; 7Vietnam National University, Ho Chi Minh City 700000, Vietnam

**Keywords:** distribution, genetic marker, mahseer, morphology, Southeast Asia, *Tor*

## Abstract

**Simple Summary:**

The mahseer, particularly the *Tor* species, is of significant cultural and economic importance as a high-value freshwater food fish in the Southeast (SE) Asian region. However, overexploitation of natural stocks because of high demand and their deteriorating habitat has resulted in a marked decline of *Tor* species in the wild. There are 13 *Tor* species that inhabit SE Asian rivers. All these species share distinct morphology, which is the presence of the median lobe. The unique characteristics, including body color, mouth position, and number of lateral scales, distinguish one species from another. Nonetheless, the taxonomy of the *Tor* species remains unstable and confusing, with contradictory data presented by different authors from different countries for a single *Tor* species. Therefore, in this review, we have gathered data for the SE Asian *Tor* species, outlining their distribution, morphology, and genetic identification. In addition, the present review also proposes a list of valid *Tor* species in the SE Asian region. The proposed list will serve as a standard and template for improving SE Asia’s *Tor* taxonomy, enhancing the study’s continuity, and a better understanding of specific *Tor* species.

**Abstract:**

The king of rivers or mahseer comprises three genera: *Tor*, *Neolissochilus,* and *Naziritor,* under the Cyprinidae family. The *Tor* genus has been classified as the true mahseer due to the presence of a median lobe among the three genera. The *Tor* species are widely distributed across Southeast (SE) Asia, and 13 *Tor* species have been reported previously: *Tor ater, Tor dongnaiensis, Tor douronensis, Tor laterivittatus*, *Tor mosal, Tor mekongensis, Tor putitora*, *Tor sinensis, Tor soro, Tor tambra, Tor tambroides*, *Tor tor* and *Tor yingjiangensis*. However, the exact number of valid *Tor* species remains debatable. Different and unstandardized approaches of applying genetic markers in taxonomic identification and morphology variation within the same species have further widened the gap and ameliorated the instability of *Tor* species taxonomy. Therefore, synchronized and strategized research among *Tor* species researchers is urgently required to improve and fill the knowledge gap. This review is a current update of SE Asia’s *Tor* species, outlining their distribution, morphology, and genetic identification. In addition, the present review proposes that there are ten valid *Tor* species in the SE Asian region. This list will serve as a template and standard to improve the taxonomy of the SE Asian *Tor* species, which could serve as a basis to open new directions in *Tor* research.

## 1. Introduction

Mahseers, known as the king of rivers, are amongst the largest scale carp and a valuable group of freshwater fish in Asia classified into three genera: *Tor*, *Neolissochilus,* and *Naziritor* under the *Cyprinidae* family and *Cyprininae* subfamily [1]. Among these three genus, only *Tor* species are valid mahseers [1,2]. *Tor* species are attractive for sport fishing, commercially valuable as highly esteemed food, and have increasing demand for aquaculture [3]. In Southeast Asia (SE Asia), *Tor* species are among the major species captured and produced by aquaculture [4]. Unfortunately, degradation of the natural habitat caused by the construction of dams in the rivers, deforestation, agricultural development, overfishing, pollution, and a lack of policy for *Tor* species fisheries has led to near extinction of these species [3,5,6,7].

This king of the rivers is endemic to Asia, and originated from Southwest China; it is widely distributed in West Asia across the trans-Himalayan region encompassing the rivers in Pakistan, India, Sri Lanka, Bangladesh, and Myanmar. The *Tor* species’ habitats further extend to most SE Asian nations across Thailand, Laos, Vietnam, Malaysia, and Indonesia [1]. *Tor* species mainly thrive in fast-flowing large rivers and lakes, and they migrate upstream in rivers with clear water streams and pristine and rocky bottoms for breeding [8]. The role of *Tor* species in the river ecosystem remains elusive due to limited studies. Nevertheless, studies have shown that *Tor* species migrate for spawning, maintaining the nutrient balance between the river or lake and stream [9,10]. For the past five years, only a few studies have become available that describe the distribution and population density, which further contributes to the lack of information about the *Tor* species in SE Asia [11].

The word mahseer has its origins in Bengal, the two root word Maha means greatness and Seer means mouth or head [2]; literally, it means having a strong and large head. Generally, *Tor* species are characterized as large-sized freshwater fish with a compressed and elongated body [12], a strong and large cycloid scale, a large mouth, and a large tail in addition to strong muscles and fins [2]. In the previous two decades, the morphology of *Tor* species has been used for speciation. Nonetheless, due to locality, environment adaptation, and genetic variation, *Tor* species have undergone phenotypic changes [2,13]. In SE Asia alone, 13 *Tor* species have been reported [1,14,15,16,17,18,19,20,21,22,23,24,25,26,27,28,29]. Phenotypic variation has caused instability and confusion for *Tor* species speciation in the SE Asian region [13]. As a result, some *Tor* species have been re-classified into other genera or re-assigned into other *Tor* species [3]. This unstable taxonomy problem has further added the difficulties to conserve *Tor* species.

In the last ten years, DNA sequences, particularly mitochondrial DNA, have been used for *Tor* species identification and speciation as well as using morphology alone [1]. Three common mitochondrial gene sequences have been used for identification and speciation purposes [13,15,30,31]. Nonetheless, the unstandardized approach and genetic variation also create instability and confusion for *Tor* species speciation in the SE Asian region [13].

There are no reviews that collectively summarize distribution, locality, morphology, and genetic identification specifically for SE Asia’s *Tor* species. Therefore, this review aims to provide a comprehensive focus and discussion on the current updates on the distribution, locality, morphology, genetic identification, and taxonomy of SE Asia’s *Tor*. The information in this review will provide another point for the related agencies or biologists to develop strategies to conserve wild *Tor* species, maintain *Tor* aquaculture’s productivity, and preserve their habitats and surrounding biodiversity.

## 2. The Genus *Tor*

Among the three genera of mahseers, the genus *Tor* (Figure 1) has been considered the “true mahseers” based on the morphological structure of the median lobe present in this genus but not in the other two genera [1,3]. Although 24 *Tor* species were recorded previously [1,14,15,16,17,18,19,20,21,22,23,24,25,26,27,28,29], a recent report in Eschmeyer’s Catalog of Fishes 2021 (as of 1 March 2021) shows that only 17 out of 24 species are validated as *Tor* species (Table 1) [32]. A recent review by Pinder et al. (2019) has concluded that there are 16 valid species of *Tor,* and all these species have been listed in the IUCN Red List of Threatened Species^TM^ (Version 2018-2) [3]. More recent versions and data from the IUCN Red List of Threatened Species^TM^ (Version 2020-3) have shown that there are now 18 *Tor* species listed as endangered species [11]. In SE Asian countries, the presence of 13 *Tor* species was reported previously (Table 2) [1,3,15,17,26,28,33,34,35,36,37]. From these 13 reported *Tor* species, only ten species are considered valid by two databases, Eschmeyer’s Catalog of Fishes 2021 and a review by Pinder et al. (2019), leaving another three species invalid or their identity remaining uncertain (Table 2) [3,32]. Furthermore, all ten valid *Tor* species were listed in the IUCN Red List of Threatened Species^TM^ (Version 2020-3). Amongst the listed SE Asian *Tor* species, there were two species classified under “vulnerable” and “endangered” categories, two species classified under “near threatened,” and six species classified under “data deficient” (Table 2). In addition, out of these ten listed *Tor* species, three species show a decreasing trend in their population number, and seven species are of unknown status [11].

## 3. Geographical Distribution of *Tor* Species in the SE Asian Region

### 3.1. Malaysia

Peninsular Malaysia is home to two *Tor* species, *T. tambroides* and *T. tambra* (see Table 2), where they inhabit around 30 rivers [2]. In Peninsular Malaysia, the *Tor* species are called “Kelah”. Most of the *Tor* species in Peninsular Malaysia are found upstream of the main river basins, which are the surrounding areas of rivers that have remained conserved and untouched by development activities.

On the west coast of Malaysia, Ulu Muda River in the state of Kedah serves as home to a large population of *Tor* species. Other rivers in the state of Kedah inhabited by the *Tor* species include Telian and Gawi Rivers [2]. Meanwhile, in the state of Perak, it has been reported that the *Tor* species inhabit Perak, Bernam, Karawang, Kejar, Mangga, Sungkai, Sara, Sigor, and Tiang Rivers and also Temenggor Lake (Figure 2) [2]. Selangor is the most advanced and rapidly developing state in Malaysia, with a large population. Therefore, deforestation to accommodate rapid development and the growing population have resulted in the loss of rivers [41]. Furthermore, the rapid growth of industry and housing estates near existing rivers has caused pollution to the rivers [42]. Hence, *Tor* species have been reported only in Kancing, Langat, Semenyih, and Selangor Rivers [2]. In the state of Negeri Sembilan, the *Tor* species have been reported to inhabit Kampung Esok [43] and Serting Rivers [44].

On the east coast of Malaysia, the *Tor* species have been reported to inhabit Galas, Nenggiri, and Lebir Rivers that meet to form the Kelantan River in the state of Kelantan. However, a study in 2015 described the failure to capture the *Tor* species, especially *T. tambroides* in the Kelantan River [45,46]. From 2016 to date, there no study has been conducted to validate the disappearance of *Tor* species in the Kelantan River. Further, there is evidence that the *Tor* species inhabit Pergau Lake, Jeli Province, in the state of Kelantan [47], Kenyir Lake, and Petuang River [2] in the state of Terengganu.

A recent study has shown that the population of *T. tambra* in Kenyir Lake is decreasing at an alarming rate [57]. There are reports of the presence of *T. tambroides* and *T. tambra* in the Keniyam River in the National Park of Pahang and in the Pahang River, which is the longest river in Peninsular Malaysia as well as in Tembeling, Jelai, Lipis, Rompin, and Merchong Rivers in the state of Pahang [2]. Although it has been reported that rivers (Endau, Kinchin, Kemapan, Jasin, and Marong) in the state of Johor, the southern state of Peninsular Malaysia, are inhabited by the *Tor* species [2], recently it has been reported that the *Tor* population is limited only to the Endau River [58].

The Malaysian states of Sabah and Sarawak, on the island of Borneo, have large areas of untouched and conserved forest. The *Tor* species in Sabah are locally known as “Pelian”, while in Sarawak, the *Tor* species are known as “Empurau”, meaning “unforgettable taste” in Chinese characters. In the state of Sabah, *T. tambra* and *T. tambroides* are the most common species that inhabit the Moyog and Tenom Rivers [2]. On the other hand, in the state of Sarawak, the most common *Tor* species are *T. douronensis* and *T. tambroides*; *T. douronensis* has been reclassified as a synonym of *T. tambra* [3]. Phylogenetic analysis by Walton et al. (2017) showed that *T. tambra* found in Borneo is genetically different from *T. tambra* and *T. tambroides* from Indonesia and Peninsular Malaysia, which might be a new species of *Tor* that remains to be confirmed [13]. Generally, the *Tor* species in the state of Sarawak can be found in most major rivers, such as Trusan, Limbang, Rejang, Layar, Sadong, Sarawak, and Batang Lupar Rivers [51]. However, the exact number of *Tor* species in Sarawak remains elusive because, among the 22 river basins in Sarawak, many remain unexplored.

### 3.2. Thailand

In Thailand, the *Tor* species are less popular as a food, or protein source than other large indigenous species from the same family, such as *Probarbus jullieni* and *Catlocarpio siamensis* [59]. Thus, fewer studies have been conducted, and there are a limited number of publications related to the *Tor* species from Thailand. The information in the following paragraph is from previous studies done over a decade ago and there are limited recent studies on these species in Thailand. *Tor* species that have been reported to inhabit lakes and rivers of Thailand include *T. tambroides* [1], *T. tambra, T. sinensis*, *T. douronensis*, and *T. putitora* [50]. Amongst these species, *T. tambroides* is widely distributed in Thailand [35,50,60]. Nevertheless, *T. tambroides* has gained much interest as a high-value food fish and ornamental fish among the Thai people [59,61,62]. This has accelerated the exploitation of wild fish both as broodstock for aquaculture and a premium supply for the ornamental fish market [63]. Moreover, habitat alteration (e.g., damming or deforestation) and deterioration have been additional threats to these valuable species [64]. Therefore, it is urgent to gather information on species distribution and diversity to enable proper conservation plans before some species are lost.

There are 25 river basins in Thailand (254 sub-basins), among which the two largest rivers are the Chao Phraya and Mekong [65]. The Mekong runs from China through Myanmar and Laos, and its tributaries are home to some of the world’s most diverse species of freshwater fishes, including the *Tor* species [35,52,66,67]. The Chao Phraya and its tributaries occupy Thailand’s central plains. It emerges from four rivers, Ping, Wang, Yom, and Nan that merge in Nakorn Sawan and continue flowing southward until it reaches the Gulf of Thailand in Samut Prakarn, south of Bangkok. *T. tambroides* was reported to inhabit the Chao Phraya and its tributaries [49,68]. Besides the two major rivers, in the western part of the country, Mea Klong River is important and runs in a north to south direction from Kanchanaburi Province and reaches the Gulf of Thailand in Samut Sakorn Province, south of Bangkok. Mae Klong has been recorded as being home to *Tor* species (see Figure 2) [1,50].

In the northwestern part of Thailand, *T. tambroides* have been reported to inhabit rivers in Chiang Mai and Mae Hong Son Provinces [35]. Meanwhile, northeastern part of Thailand, from Nong Khai to Bueng Kan, Nakhon Phanom, and Kaeng Tana National Park, serves as a river basin for the 10th longest river in the world, the Mekong River, which is also home to *T. tambroides* [50]. In addition, *T. tambroides* are found to inhabit Phetchaburi Basin located in Phetchaburi Province, northwest of the Gulf of Thailand [60]. The southern region of Thailand, located on the Thailand peninsula, stretches from Kra Isthmus to the Malaysian border. There are 12 rivers in this area; among them, the Tapi River is the longest in the high mountain ranges of the southern peninsula and is home to 319 fish species, including *T. tambroides*, *T. tambra*, and *T. douronensis* [68]. The *T. tambroides* have been reported to inhabit Cheow Lan Reservoir, Khao Sok National Park in Surat Thani Province [35]. Despite its debatable taxonomic classification, *T. douronensis* has been reported in peninsular Thailand, including in the Chao Phraya and Mekong Basins. Furthermore, this species has also been reported in Si Sawat District, Kanchanaburi Province (central Thailand), where lies the Srinakarin Reservoir, rivers in Chiang Mai Province (northern Thailand), and also Pattani River in the south [35].

### 3.3. Vietnam

Only a few studies describe the presence of *Tor* species in Vietnam, which mainly focus on the Central Highlands in the southern region. Several *Tor* species have been reported to inhabit the highland riverine systems in Vietnam, including *T. tambra*, *T. tambroides*, *T. sinensis*, *T. mekongensis,* and *T. dongnaiensis* [15,69]. The study conducted by Hoang et al. (2015) has shown *T. mekongensis* and *T. sinensis* in Krong No River (see Figure 2) [15]. This river rises from Chu Yang Sin Mountain (>2000 m) and flows west 156 km, and meets River Krong Ana before connecting to Sre Pok River, a transboundary tributary of the Mekong River [70]. In addition, Hoang et al. (2015) have also shown the presence of *T. dongnaiensis* in the middle Dong Nai River, Cat Tien National Park, and in Lam Dong Province [15]. Dong Nai River is the longest inland river in Vietnam, at 586 km, starting from Lam Dong Highlands and flowing to the East Sea. A previous study by Mai et al. (1992) has reported *T. tambroides* in the middle and upstream of Dong Nai River. However, Hoang et al. (2015) suggested that *T. tambroides* described by Mai et al. (1992) is *T. dongnaiensis* based on morphological characteristics. Therefore, the natural distribution of *T. tambroides* in Vietnam is still in question [3,15].

In another study, Phan and Duong (2014) collected *T. tambroides* from some locations (Dak Lak, Dak Nong, and Lam Dong) of the Central Highlands where Sre Pok River flows which belongs to the Mekong River Basin [71]. However, the validity of *T. tambroides* from these areas remains questionable as the analysis of this species showed it belongs to *T. sinensis*. Given that the two species *T. dongnaiensis* and *T. tambroides* are present in separate river basins (Dong Nai and Mekong, respectively) in Vietnam and the latter is found in Laos sharing the Sesan, Sre Pok, and Sekong (3S) River Basins with Vietnam, their distribution and classification should be further investigated. Besides *T. tambroides*, the distribution of *T. mekongensis* in Vietnam is also questionable. Further genetic analysis of *T. mekongensis* done by Walton et al. (2017) has shown that this *Tor* species belongs to *T. tambra* [13]. Another issue about the validity of Vietnam’s *Tor* species is *T. dongnaiensis*. The genetic analysis, done by Hoang et al. (2015) to compare Vietnam’s *Tor* species with *T. tambra* (previously described as *T. douronensis*) from the Malaysian Borneo, should be revised because the *Tor* species from Malaysian Borneo did not belong to any known valid *Tor* species in SE Asian countries [13,15].

### 3.4. Myanmar

Most reports that mention the presence of *Tor* species in Myanmar are those from the Thanlwin River. This river originates from the Himalayan Plateau and passes through China, Thailand, and Myanmar, and flows into the Andaman Sea. Locally, the *Tor* species are called Nga-dauk. Several *Tor* species have been reported to inhabit the Thanlwin River (see Figure 2), which are *T. putitora* (golden mahseer), *T. tambroides,* and *T. tor* [18]. Besides these, *T. yingjiangensis* has been reported to inhabit the Mali Hka [26] and Irrawaddy [28] Rivers. Although Myanmar is well known as the primary source of fish globally, there is no recent study investigating the availability and population size of *Tor* species in Myanmar. The limited number of studies and no recent reports (in the last three years) on the *Tor* species raise concern over the number of *Tor* species which exist in Myanmar.

### 3.5. Lao PDR

Geographically, Lao PDR is the only land-locked SE Asian country. Therefore, the fish industry in Lao PDR is solely dependent on freshwater fish that come from rivers, reservoirs, ponds, and lakes [72]. A previous study in 1999 found the presence of *T. ater, T. sinensis*, and *T. tambra* that inhabit the Nam Theun/Kading River [36]. In 2011, a study showed that Sekong (Xe kong) River tributaries and Se Kaman/Xe Kaman River are home to several *Tor* species, which are *T. laterivittatus, T. tambra*, and *T. tambroides* (see Figure 2) [16]. To date, there are no other studies that have reported the population sizes of *Tor* species in Lao PDR.

### 3.6. Cambodia

Freshwater fish are captured and mostly consumed in Cambodia [73]. Nonetheless, studies on *Tor* species’ population sizes are unavailable. Previous studies have shown that *T. tambroides, T. tambra*, and *T. sinensis* inhabit the Mekong River in Cambodia [66,74]. Unfortunately, there is no study to show the presence of *Tor* species in other rivers of Cambodia or even in Tonle Sap Lake.

### 3.7. Indonesia

Indonesia is the largest island country and separated from mainland SE Asia by sea. Nonetheless, Indonesia somehow shares similar *Tor* species with some SE Asian mainland countries such as Malaysia, Thailand, Vietnam, and Cambodia. The *Tor* species reported in Indonesia’s rivers are *T. tambroides*, *T. tambra,* and *T. douronensis* [19]. The wide distribution of *Tor* species across SE Asia’s mainland, Borneo, Sumatra, and Java of Indonesia could be because the Mekong River system was connected with these islands’ river systems during the Pleistocene, known as the Sunda Shelf [75]. After the rising of sea levels, SE Asia separated from the Indonesian islands, as it is now.

The *Tor* species are locally known as Keureling fish. Starting from Sumatra, four *Tor* species, *T. tambroides*, *T. douronensis*, *T. tambra*, and *T. soro*, have been recorded as inhabitants of this area [76]. Currently, *T. soro* is classified as *N. soro* [77]. In Sumatra, several studies have reported the presence of *Tor* species. A recent study showed *T. tambroides* in the upstream of Wampu waters, northern region of Sumatra [19]. Another study also showed *T. tambroides* in the Manna and Tarusan Rivers of the western part of Sumatra and Bahorok River and Berkail River. Meanwhile, a survey conducted in the Batang Toru river system, South Tapanuli, North Sumatra, has shown several *Tor* species, namely *T. tambra*, *T. douronensis*, and *T. tambroides* (see Figure 2) [53]. Furthermore, a study in the Kreuang Sabee River, Aceh Jaya District, Indonesia showed only *T. tambra* [78]. Although *T. douronensis* has been classified as an invalid species, this species has been widely studied in Indonesia and found widely distributed in several rivers on Sumatra, namely Batang Gumanti River, Batang Antokan River, Batang Malalo River, Batang Matur River, Batang Sinuruik River, and Lubuk Mangkuih River [79].

Moving toward south of Sumatra, Java is also home to the *Tor* species [80]. A recent study documented *T. tambroides* in the rivers of West Java, particularly in Cimanuk River that flows from Pappandayan Mountain into the Java Sea [54]. Another recent study reported that *T. douronensis* inhabits riverine systems surrounding Cereme Mountain located in Kuningan District, West Java [81]. The river system areas inhibited by *T. douronensis* are Balong Dalem, Darma Loka, Cigugur, Cibulan, and Bale Kambang [81]. In Kalimantan of Borneo, *T. tambroides* have been reported in riverine systems of Muller Mountain in Central Kalimantan [55,56]. In addition, *T. tambroides* and *T. tambra* have been reported in the Mendalam river system, Betung Kerihun National Park, West Kalimantan. As compared to Sumatra, studies on *Tor* species on Java and Kalimantan are limited. However, in Bogor, South Jakarta, Java, the Research Institute for Freshwater and Fisheries Extension is actively engaged in *T. douronensis* and *T. tambroides* aquaculture and preservation research [82].

### 3.8. Brunei, Singapore, Timor-Leste, and the Philippines

A recent study in 2018 reported that *T. tambroides* and *T. tambra* inhabit the Brunei riverine system of the Temburong River [83]. Since then, no other *Tor* species study has been reported from Brunei. Singapore has no recent studies on *Tor* species. *T. tambroides* was reported in Singapore riverine systems back in 1966 [84]. However, in 1997, *T. tambroides* were classified as extinct in Singapore [85]. The Philippines and Timor-Leste were not part of the Sunda Shelf during the Pleistocene [86]. Therefore, the riverine systems in these two countries are not connected with any of the mainland of SE Asia; hence, *Tor* species have not been reported in the Philippines and Timor-Leste.

## 4. The Morphological Features of SE Asia’s *Tor* Species

Morphological features such as shape, size, fin ray, and median length have been used previously to distinguish *Tor* species. However, this methodology has caused confusion and debate among taxonomists [13]. The *Tor* species’ morphological features show high intra- and interspecies variability that is influenced by environmental factors and localities [2,13]. This has resulted in difficulty to validate the *Tor* species in SE Asian countries as well as in generating stable databases for *Tor* species. Therefore, establishing a standard morphological identification, morphometric measurement, and genetic identity analysis will prevent data variation and avoid confusion. Therefore, this review has summarized the SE Asian *Tor* species’ distinct morphological characters, which could help towards *Tor* conservation.

### 4.1. T. ater (Robert, 1999)

*T. ater* is the smallest of the SE Asian *Tor* species. Generally, *T*. *ater* is characterized by having a short median lobe, longitudinal dark stripe on the body, and all fins have a dark or black color (Table 3) [14].

### 4.2. T. dongnaiensis (Hoang et al., 2015)

*T. dongnaiensis* has morphological characters that distinguish it from the Indonesian *T. tambroides*, namely, yellow to gray body color. The morphological feature of the mouth is the absence of a median projection of the upper lips, and unequal caudal fin [15]. Generally, the appearance of this species includes conical head shape, pointed snout, subterminal mouth with fleshy lips, long lower lip of median lobe, and straight and pointed rostral hood (see Table 3). However, genetic analysis done by Walton et al. (2017) shows no clear separation between *T. dongnaiensis* studied by Hoang et al. (2015) and *T. tambra* from West Java and Peninsular Malaysia [13]. Therefore, this species is postulated to be classified under *T. tambra* [13]. Nonetheless, a recent review by Pinder et al. (2019) suggests that *T. dongnaiensis* remains as a valid *Tor* species [3]. Therefore, a large-scale study is required to validate this species’ uncertainty.

### 4.3. T. douronensis (Valenciennes, 1842)

The taxonomy of *T. douronensis* is the most complicated amongst the *Tor* species. This name (*T. douronensis)* has been debated since 1999 and continues to date [3,13,32]. Several researchers, including Roberts (1999) and Kottelat (2013), have debated the validity of *T. douronensis* as there is no specific morphological feature that can distinguish between this species and *T. tambra* [36,52]. Hence, Kottelat (2013) has postulated that *T. douronensis* found in Peninsular Malaysia and Borneo (Sabah and Sarawak) is a synonym of *T. tambra* [52]. Despite the unstable taxonomy, researchers in Indonesia continue to use *T. douronensis* for species identification instead of *T. tambra* [94,95,96,97]. This postulation has been supported by Rahayu et al. (2015) as their study reported that *T. tambroides* and *T. douronensis* are distinct species through genetic classification, along with morphological classification [98]. Morphologically, *T. douronensis* has a short lower median lobe, the absence of an upper median projection, and a blunt rostral hood (see Table 3). Additionally, the color of *T. douronensis* is silvery with a darkish back. Haryono and Tjakrawidjaja (2006) showed that three *Tor* species, *T. tambroides*, *T. douronensis* and *T. tambra,* from the same source are distinguishable from each other through their body measurements [93]. From the study mentioned above, the body measurements that differentiate the *Tor* species are interorbital width, caudal peduncle length, caudal peduncle depth, head width, and body depth [93]. Interestingly, *T. douronensis* is different from *T. tambra* with its less than 10 mm interorbital width and large caudal peduncle. Additionally, genetic analysis done by Walton et al. (2017) showed that there is a clear separation in the phylogenetic tree between this species and *T. tambra* from West Java, Indonesia, and from Peninsular Malaysia [13]. Therefore, the authors suggest that the *Tor* species from Borneo might be a new *Tor* species. However, for further confirmation of this species’ validity, morphological and genetic comparisons between *T. tambra* and *T. douronensis* from the Mekong riverine system, Peninsular Malaysia, Borneo, Sumatra, and Java should be performed to resolve this varying taxonomy.

### 4.4. T. laterivittatus (Zhou & Cui, 1996)

This species’ unique characteristics are its elongated median lobe of the lower lip, the upper lip is rolled upward and backward, the presence of a longitudinal stripe along the body, and a profoundly concave dorsal (see Table 3) [37]. Amongst the *Tor* species., *T. laterivittatus* is the least studied, and therefore information on this species is mainly from fishers’ local knowledge [3].

### 4.5. T. mekongensis (Hoang et al., 2015)

From the word “Mekong”, it is evident that *T. mekongensis* inhabit the Mekong River system [15]. The general appearance of *T. mekongensis* is as follows: longer head, blunt snout, blunt rostral hood, subterminal mouth with fleshy lips, and a short median lobe of the lower lip (see Table 3). A study by Hoang et al. (2015) has reported that this species is a valid *Tor* species. Genetic analysis of *T. dongnaiensis* (Vietnam) and *T. tambra* (previously known as *T. douronensis* from Borneo (Malaysia)) shows a clear separation of these species’ phylogenetic tree. In contrast, genetic analysis of *T. mekongensis* [15] and *T. tambra* from West Java, Indonesia, and Peninsular Malaysia shows no difference between the two species along the phylogenetic tree [13]. Furthermore, a recent review has also suggested that *T. mekongensis* is a synonym of *T. dongnaiensis* [3]. This species’ variable taxonomy requires a new study to review the validity of all the *Tor* species in SE Asian countries.

### 4.6. T. mosal (Hamilton, 1822)

*T. mosal* is considered a synonym of *T. putitora* [99]. *T. mosal is* also known as the copper mahseer due to its near reddish coloration on the anal and pectoral fin, which resembles *T. putitora* [3]. However, the head length of this species is distinctively shorter than *T. putitora* (see Table 3) [100]. Furthermore, genetic analysis has shown that *T. mosal* is different from *T. barakae*, *T. putitora*, and *T. tor* [100].

### 4.7. T. putitora (Hamilton, 1822)

In India, *T. putitora* is known as the golden mahseer. The color of this species appears greenish and silvery on the side of the body, which then turns reddish yellow or golden on the anal and pectoral fin that gives it the name (see Table 3) [101]. Like *T. tor,* which originated from the Himalayan Plateau, this species is endemic to the Myanmar riverine system [7]. This species’ unique characteristics are an elongated and nearly straight body, a small mouth with lower jaw slightly shorter than the upper jaw, and a deeply forked caudal fin.

### 4.8. T. sinensis (Wu, 1977)

*T. sinensis* is another *Tor* species that predominantly inhabits the Mekong River system [67]. This species has unique characteristics that differ from other sympatric *Tor* species. Compared to *T. ater*, *T. sinensis* is characterized by fewer number of a lateral, predorsal and transverse row scales, a long median lobe on the lower lip, and the absence of stripes along the body (see Table 3) [15]. *T. sinensis* has similar features to *T. tambroides*; a large standard length over body depth ratio and the presence of a median projection of the upper lip (see Table 3). Compared with *T. mekongensis*, *T. sinensis* uniquely has a long rostral hood and many lateral scales (23–28). The color of *T. sinensis* is usually dark on the head and back with a silver gray or yellowish color along the body. Furthermore, unlike other *Tor* species, *T. sinensis* has a longitudinal stripe stretching from the head to the caudal fin.

### 4.9. T. tambra (Valenciennes, 1842)

Like *T. tambroides*, *T. tambra* can be found in many riverine systems of the SE Asian countries, including the Mekong River system, Thailand, Peninsular Malaysia, Sumatra, Java, and Borneo [13]. The color of this species varies depending on the locality, including reddish, olive, dark, or slightly olive [13,15]. The morphological features that differentiate *T. tambra* from *T. tambroides* are the lower median lobe’s varying size and the blunt rostral hood (see Table 3). Other features that *T. tambra* shares closely with other *Tor* species are the absence of an upper median projection and an equal caudal fin [13].

### 4.10. T. tambroides (Bleeker, 1854)

The body color of *T. tambroides* is silver/bronze/reddish. The unique morphological feature that specifically represents *T. tambroides* is the presence of an upper median projection (see Table 3) [93]. Other features of *T. tambroides* common among other *Tor* species include sub-terminal mouth position, long lower median lobe, pointed rostrum hood, and an equal caudal fin lobe [13,15].

### 4.11. T. tor (Hamilton, 1822)

The prominent characteristics that differentiate *T. tor* from other *Tor* species in Myanmar is the big head, large scales, sub-terminal mouth with an interrupted fold of the lower lip, two pairs of large barbels, and lateral line scales (see Table 3) [15].

### 4.12. T. yingjiangensis (Chen and Yang, 2004)

*T. yingjiangensis* was first identified by Chen and Yang (2004), and originated from Yunnan Province, China [28]. Briefly, *T. yingjiangensis* uniquely have a longer, conical head shape and a slightly convex body (see Table 3). The body color of *T. yingjiangensis* is yellowish and black or light brown in the middle. All fins are yellowish, and there is no mid-lateral line [28].

## 5. Genetic Identification of SE Asia’s *Tor* Species

Conventional taxonomy relies solely on morphological characters. However, it is known that phenotypic plasticity occurs in animals due to adaptation to environmental conditions [3,15,36,102]. In addition, in specific cases, variable gene expression is also observed, e.g., the saddle-back trait of Nile tilapia shows a wide range of phenotypes, from a standard dorsal fin to the lack of a dorsal fin, despite having the same genotype [103]. Phenotypic variations within the *Tor* species are other factors of taxonomic confusion [3,104]. Recent studies have employed molecular markers to solve taxonomic ambiguity [13,15,105] and/or assess the genetic diversity of *Tor* [43,106,107,108]. Most of the studies have employed mitochondrial genes, while others have used nuclear DNA markers, such as microsatellites, for species identification. This review focuses on two types of genetic markers and their application for *Tor* studies.

### 5.1. Mitochondrial DNA

Generally, animal mitochondrial DNA (mtDNA) is 16–17 kb in size and consists of two rRNA-encoding, 22 tRNA-encoding, and 13 protein-encoding genes, including NADH-ubiquinone oxidoreductase chain 1–6 (ND1–6), NADH-ubiquinone oxidoreductase chain 4L (ND4L), cytochrome b (Cytb), cytochrome c oxidase subunit I-III (COX 1–3), and lastly complex V subunits of ATPase 6 (ATP6) and 8 (ATP8) [109]. The characteristics of mtDNA are uniparental inheritance through maternal lineage, non-recombination, and higher mutation rates as compared with nuclear DNA [110]. With these characteristics, mtDNA has been used extensively for phylogenetic analyses for species identification and specification. For more than two decades, the identification and specification of *Tor* species has remained unstable. In this regard, different mtDNA gene sequences have been used for species identification. The most frequently used is cytochrome oxidase I (*CO1* or *COX1*) [13,15,44,107,111] followed by cytochrome b (*Cyt b*) and ATPase subunits 6 and 8 (*ATPase 6/8*) [1,106,108]. The other mitochondrial gene used in *Tor* studies is the *16S rRNA* [1]. Complete mitochondrion genome sequences have been reported for a few *Tor* species, such as *T. tambroides* [112] and *T. tor* [104].

In this review, we performed phylogenetic analyses using *COX1*, *Cyt b,* and *16S rRNA* gene information databases from the National Center for Biotechnology Information (NCBI) [113]. Four phylogenetic analyses were performed using MEGA X software (Version 10.1) [114,115]. First, we performed the phylogenetic analysis for the complete sequence of the *COX1* gene. In this analysis, seven complete sequences of COX1 of *Tor* species samples were selected, which are *T. douronensis, T. putitora, T. sinensis, T. tambra, T. tambroides,* and *T. tor* (Appendix A). These seven complete sequences of *COX1* of *Tor* species were aligned together with five selected complete *COX1* sequences of *Neolissochilus* species, *N. benasi*, *N. hexagonolepis*, *N. hexastichus*, *N. soroides,* and *N. stracheyi*, which acted as the out-group (see Appendix A) [112,116,117,118,119,120,121,122,123,124,125]. The phylogenetic trees were constructed using the maximum likelihood method and Tamura–Nei model [126]. All the samples used in the first phylogenetic analysis were used as reference samples for the subsequent phylogenetic analyses. Second, we performed the phylogenetic analysis of complete and partial/short sequences of the *COX1* gene. For this phylogenetic analysis, we extended the first phylogenetic analysis by adding seventy-one short/partial sequences of *COX1* of *Tor* species obtained from NCBI databases (Appendix A) [13,15,107,127,128,129,130,131,132,133,134,135,136,137,138,139,140,141,142]. Then, the phylogenetic trees were constructed using the method and model used in the first phylogenetic analysis. Similar to the second phylogenetic analysis, the third and fourth phylogenetic analyses were performed using complete and partial/short sequences of *Cyt b* and *16S rRNA* genes. Briefly, twelve complete sequences of *Cyt b* or *16S rRNA* genes from similar *Tor* and *Neolissochilus* samples used in the first phylogenetic analysis were obtained from NCBI databases. Then, these complete sequences were aligned with a short/partial sequence of *Cyt b* or *16S rRNA* genes of selected *Tor* species obtained from NCBI databases. For *Cyt b* phylogenetic analysis, eighty-three short/partial sequences of *Cyt b* of *Tor* species were selected (Appendix A) [1,108,112,116,117,118,119,120,121,122,123,124,125,127,142,143,144,145,146,147,148,149,150,151,152,153,154], while for *16S rRNA* phylogenetic analysis, sixty-eight short/partial sequences of *16S rRNA* of *Tor* species were selected (Appendix A) [1,31,112,116,117,118,119,120,121,122,123,124,125,127,154,155,156,157,158,159,160,161,162,163,164,165,166]. The results of these four phylogenetic analyses are described and discussed in the following sub-section.

#### 5.1.1. Cytochrome c Oxidase Subunit I (*COX1*) Gene

Cytochrome c oxidase subunit I (*COX1*) is a gene that encodes for one of three subunits of the cytochrome c oxidase complex. The *COX1* gene has been proposed as a powerful universal marker for vertebrate species identification and specification, including the *Tor* species [167,168]. A study by Hoang et al. (2015), covering 17 *Tor* species, found that most of the morphological measurements were in agreement with the results of *COX1*, which enabled them to validate a new species, *T. dongnaiensis* [15]. Furthermore, *COX1* sequences confirmed the difference between *T. mekongensis* and *T. tambra* despite their morphological similarity. Without morphological data, Esa et al. reported the difference between *T. tambroides* and *T. douronensis* based on *COX1* sequences [44,111]. A group of unique haplotypes was observed and subsequently proposed as a cryptic lineage of *T. douronensis* [111]. *COX1* has also been used to study the genetic diversity of *T. tambroides* from Sumatra, Indonesia [107]; due to the highly conserved nature of the marker, very low genetic diversity was reported.

From our first phylogenetic analysis for complete sequences of the *COX1* gene of *Tor* and *Neolissochilus*, the phylogenetic tree constructed demonstrates that *T. tambroides* sample JX444718.1 was clustered together with *T. tambra* sample KJ880044.1 and closely related to sample AP011372.1 (presumed as *T. tambroides*) (Figure 3), while *T. douronensis*, *T. putitora*, *T. sinensis,* and *T. tor* are not clustered together. Furthermore, all listed *Neolissochilus* samples were separated from *Tor* species. In our second phylogenetic analysis, the phylogenetic tree demonstrates that *T. douronensis, T. sinensis, T. tambra,* and *T. tambroides* are distinct species (Figure 4). Similar to Walton et al. (2017), our present phylogenetic tree analyses show that the *T. dongnaiensis* and *T. mekongenesis* samples, collected by Hoang et al. (2015), and the sample KT001033.1, collected in Malaysia (presumed as *T. tambroides*), are clustered together with *T. tambra*, collected in Indonesia and Malaysia (see Figure 4) [13,15]. In contrast with findings by Kottelat (2013) and Robert (1993; 1999), our phylogenetic tree of complete and partial/short sequences of *COX1* demonstrates that *T. douronensis* is not closely related to *T. tambra* and *T. tambroides* [36,52,80].

#### 5.1.2. Cytochrome b (Cyt b) and ATPase 6/8 Gene

Besides *COX1*, the *Cyt b* gene has also been used as a reliable and useful universal marker for fish identification and specification [169]. The length of the *Cyt b* gene is about 1143 bp with low variability. However, a longer sequence is postulated to increase or improve species identification and specification [170]. Other mitochondrial gene sequences used as universal markers are *ATPase 6* and *ATPase 8* genes (referred to as *ATPase6/8*), located next to each other and overlapping by six base pairs [108]. This gene sequence is characterized by its high mutation rate, and is suitable to distinguish species with high haplotype diversity [171]. Identification of the *Tor* species, based on *Cyt b*, *ATPase6/8,* and *16S rRNA*, by Nguyen et al. (2008), found two distinct lineages in samples of *T. douronensis* from the Mekong River Basin, Sumatra, and Borneo. They suggested that the samples from the Mekong River Basin may be *T. tambra* [1]. Sati et al. (2013) and Sah et al. (2020) used *Cyt b* and *ATPase6/8* to distinguish between *T. putitora* and *T. tor* [106,108].

A phylogenetic tree was constructed by aligning the partial/short sequences of the *Cyt b* gene (*T. douronensis, T. putitora, T. sinensis, T. tambra, T. tambroides, and T. tor*) with twelve complete sequences of the *Cyt b* gene of *Tor* and *Neolissochilus* species as a reference (Figure 5). The phylogenetic tree of *Cyt b* shows a similar pattern to the *COX1* phylogenetic tree, where *T. douronensis*, *T. putitora*, *T. sinensis, T. tor,* and *Neolissochilus* sp. are separated from each other (see Figure 5). *T. tambra* and *T. tambroides* are clustered together but separated from other listed *Tor* and *Neolissochilus* species. These *T. tambra* and *Tor* samples presumed as *T. tambroides* are from Peninsular Malaysia and Sumatra, Indonesia. This finding shows that the *T. tambroides* in Malaysia require a re-visit and supports Walton et al. (2017) [13]. Notably, the present phylogenetic analysis indicates that the *T. douronensis* are grouped into two clusters. Some of the *T. douronensis*, clustered together with *T. sinensis,* are from China and Vietnam. Hence, the presumed *T. douronensis* could be *T. sinensis*. Furthermore, the *T. douronensis* from Indonesia, and the two *T. tambroides* (DQ464985.1 and EF588065.1) from Vietnam and Malaysia are closely related to *Neolissochilus.*

#### 5.1.3. 16S rRNA Gene

Previous studies have shown that the *16S rRNA* gene sequence is also a useful marker for species identification [167]. Despite its highly conserved nature, this sequence also shows polymorphism due to a high level of deletions and insertions [172]. Nguyen et al. (2006) used *16S rRNA* as a marker for identifying *T. tambroides* and *T. douronensis* in the wild broodstock collected from Borneo (Sarawak) [31]. Their results emphasized the importance of supporting data from molecular markers for taxonomic identification as they observed two genetically distinct lineages within *T. douronensis* harboring the same morphological characteristics. Furthermore, interspecific hybridization has been raised as one possible reason for the presence of a *T. tambroides* haplotype in *T. douronensis* [31]. Even though the *16S rRNA* gene sequence has been used and classified as a universal marker, the current finding shows that this gene is less powerful than *COX1* for species identification of congener cyprinids [173].

We generated a phylogenetic tree of the *Tor* species using the *16S rRNA* gene (Figure 6). In line with phylogenetic analyses of *COX1* and *Cyt b*, phylogenetic analysis of *16S rRNA* also was able to separate *T. putitora, T. douronensis,* and *T. tor*. Meanwhile, *T. tambra* and presumed *T. tambroides* samples were clustered together, however, they were separated from other listed *Tor* and *Neolissochilus* species. In contrast with phylogenetic analysis of *Cyt b* (see Figure 5), *T. douronensis* samples were clustered together into one cluster (see Figure 6). Nonetheless, *T. douronensis* from China and Vietnam were clustered into one group separated from *T. douronensis* samples from Indonesia and Malaysia. The *T. douronensis* from China and Vietnam were clustered together with *T. sinensis*, similarly to the phylogenetic analysis of *Cyt b*. In the *16S rRNA* phylogenetic analysis, we discovered that the presumed *T. tambroides* collected in Vietnam (DQ464914.1) were clustered together with *Neolissochilus* sp., but in the phylogenetic analysis of *Cyt b*, it was clustered together with *T. douronensis* from China and Vietnam (see Figure 5). However, the presumed *T. tambroides* collected in Malaysia (EF588065.1) was clustered together with *Neolissochilus* sp. (see Figure 6). This was similar the phylogenetic analysis of *Cyt b* (see Figure 5).

The present phylogenetic analyses of *COX1, Cyt b,* and *16S rRNA* were able to distinguish listed *Tor* species. Furthermore, these phylogenetic analyses also show a different or inconsistent result for similar samples. Nonetheless, from the outcome of our phylogenetic analyses and supported by the previous review, Walton et al. (2017), we would like to postulate and suggest three things: (1) *T. douronensis* should be considered as valid, (2) *T. tambroides* should also be maintained as a valid species, however, the presumed *T. tambroides* collected in Malaysia requires revalidation as *T. tambra*, (3) *T. dongnaiensis* should be reconsidered as a synonym of *T. tambra*. In addition, we present our suggested list of valid *Tor* species specifically in the SE Asian region in Table 4. This list will serve as a template or standard for improving or revaluating SE Asia’s *Tor* species taxonomy.

### 5.2. Microsatellite DNA Marker

Microsatellites are used for population genetics studies due to their high level of polymorphisms, co-dominant nature, and Mendelian inheritance [174]. Despite various genetic studies on *Tor* species, there is no study comparing all the *Tor* species present in SE Asian countries. A microsatellite is a repetitive sequence of DNA with specific motifs repeating up to 50 times, occurring at thousands of locations within the genome. Microsatellites are also known as simple sequence repeats or simple sequence length polymorphisms. This sequence is highly susceptible to mutation compared to other DNA regions, resulting in a high level of polymorphisms. Despite their advantages (e.g., highly polymorphic, co-dominant, and Mendelian inheritance nature), microsatellite markers require prior knowledge whereby the primers used for amplification have to be developed explicitly from flanking sequences of that species (or related species) [174]. Only a few microsatellite primers have been designed for the *Tor* species, e.g., 10 polymorphic loci developed from *T. tambroides* DNA [30]. A study by Nguyen (2008) shows that, based on nine microsatellites, they gained information useful for the conservation of *T. douronensis* in seven river systems in Sarawak. The study shows that the genetic variation in each population is low. Due to the high level of polymorphisms of microsatellites, it is difficult to find the “diagnostic loci” monomorphic on different alleles in different species. There are no diagnostic microsatellite loci between *T. tambroides* and *T. douronensis*; microsatellites show a longer genetic distance between species (*d* = 0.1412 − 0.2201) than within species (0–0.0134 for *T. tambroides* and 0.0128 for *T. douronensis*) [43]. Microsatellite analysis, when used together with the *COX1* gene to compare differences between *T. tambroides* and *T. douronensis*, does not resolve the ambiguity, which suggests that microsatellite DNA is not a powerful method for species identification.

## 6. Conclusions

Collectively, there are 13 *Tor* species reported to inhabit SE Asian rivers, except in the Philippines and Timor-Leste. Nonetheless, research related to the *Tor* species, particularly in Myanmar, Lao PDR, Cambodia, and Brunei has remained scarce (fewer than five studies) and, therefore, requires more attention. Furthermore, out of 13 *Tor* species, the distribution of *T. laterivittatus* and *T. yingjiangensis* remains elusive as there are fewer than five reports and no new research has been conducted on these two species since 2011 and 2017, respectively. From this review, it is clear that limited data and the rapid decline in the *Tor* species population indicate an urgent need for collaborations within SE Asian countries because failure to recognize a distinct taxon may lead to its extinction.

Even though 13 *Tor* species have been reported, phylogenetic analysis based on genetic markers classifies *T. tambroides* from Malaysia and *T. dongnaiensis* and *T. mekongensis* from Vietnam as *T. tambra*. Furthermore, *T. douronensis* is not synonym of *T. tambra* nor *T. tambroides*. Therefore, based on phylogenetic analysis, we postulate that only ten species inhabit the SE Asian rivers. This would serve as a standard template for future comprehensive studies of SE Asia’s *Tor* taxonomy, particularly of *T. tambra*, *T. tambroides*, and *T. douronensis*. In terms of the speciation of SE Asia’s *Tor*, multiple phylogenetic analyses using sequences of different genes simultaneously and morphological analysis would enable species identification.

To date, the whole genome and transcriptome sequences of the *Tor* species are not available. Sequencing of the *Tor* species genome and transcriptome will provide a powerful tool to address questions of evolutionary biology, species identification, morphological variations, and sequences of genes related to reproduction, sex differentiation, and growth, which will be useful for conserving the *Tor* species.

## Figures and Tables

**Figure 1 biology-10-00286-f001:**
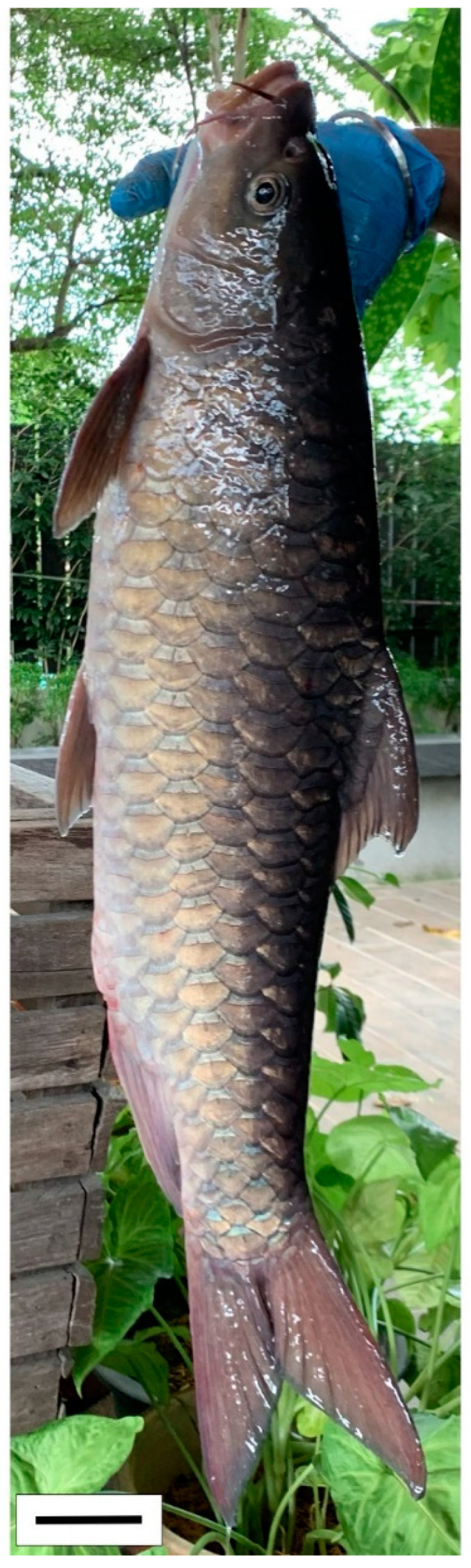
*T. tambroides* from an aquaculture farm in the state of Negeri Sembilan, Malaysia. Note: Scale bar *=* 5 cm.

**Figure 2 biology-10-00286-f002:**
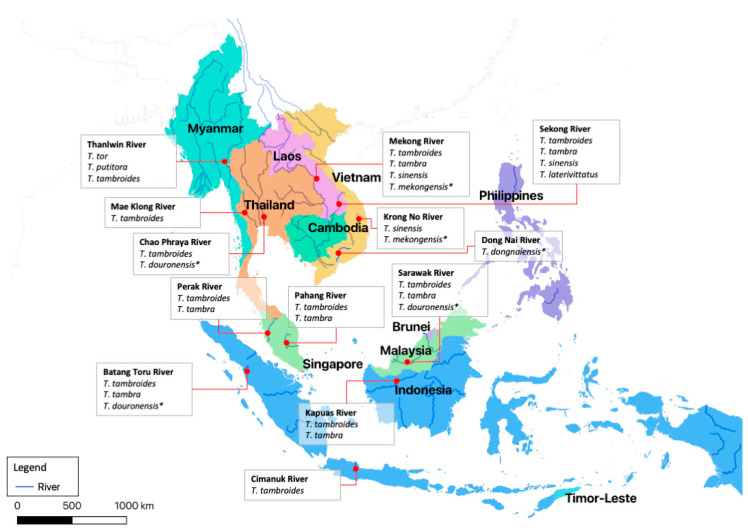
Geographical distribution of *Tor* species in the SE Asian region. Source: The GIS map was generated with QGIS software version 3.10. The map was obtained from Google Maps databases. The locations of the rivers were obtained from Google Maps and River, Lake, and Centreline QGIS databases. The distribution and locality of *Tor* species were generated based on the reports from previous ichthyofauna studies conducted in SE Asia [2,15,16,18,48,49,50,51,52,53,54,55,56]. Note: * symbol is represent uncertain species.

**Figure 3 biology-10-00286-f003:**
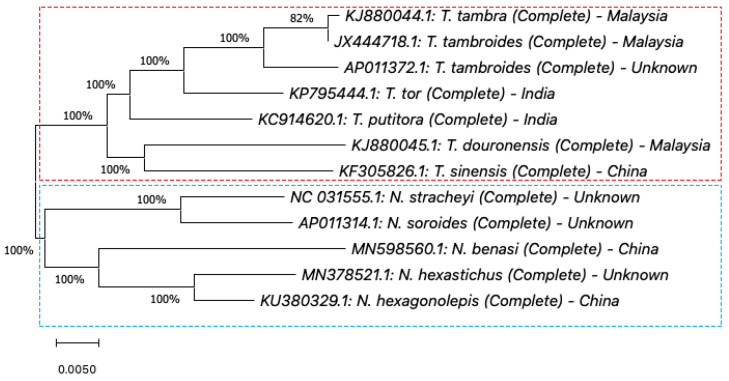
Phylogenetic tree of *Tor* species using seven complete sequences of the *COX1* gene. The evolutionary history of the listed *Tor* species was inferred using the maximum likelihood method and Tamura–Nei model [126]. These analyses were conducted using MEGA X (Version 10.1) [114,115]. Percentages on the branches represent bootstrap values.

**Figure 4 biology-10-00286-f004:**
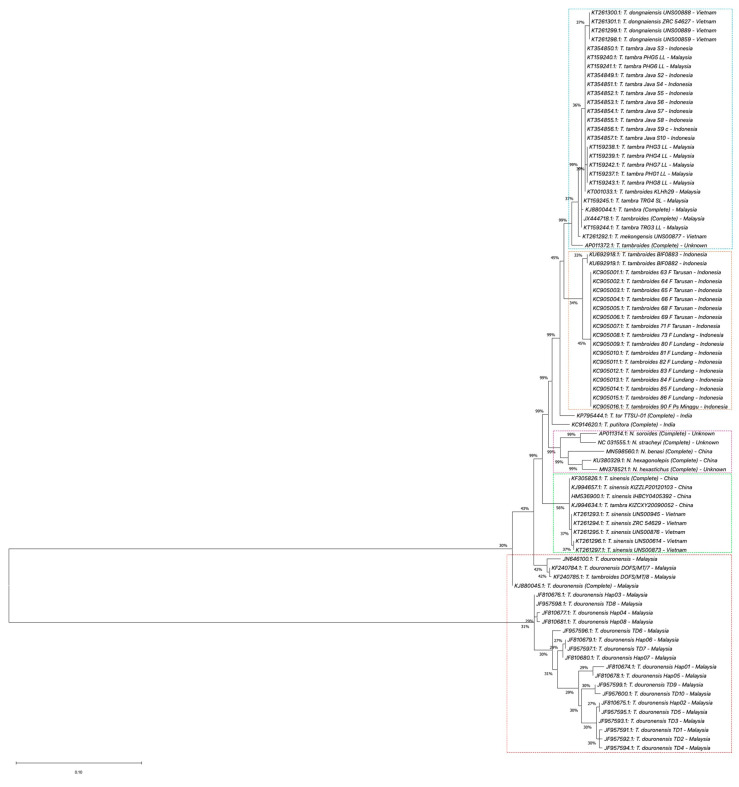
Phylogenetic tree of *Tor* species using the *COX1* gene. The evolutionary history of the listed *Tor* species was inferred by using the maximum likelihood method and Tamura–Nei model [126]. These analyses were conducted using MEGA X (Version 10.1) [114,115]. Percentages on the branches represent bootstrap values.

**Figure 5 biology-10-00286-f005:**
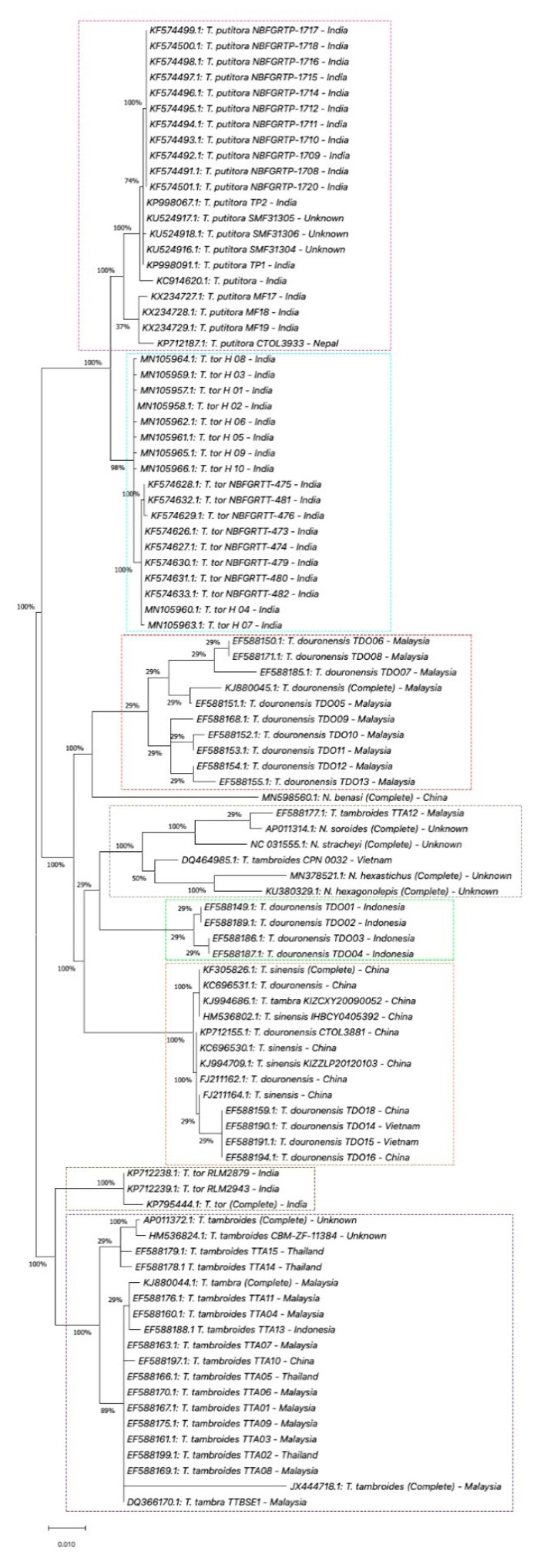
Phylogenetic tree of *Tor* species using the *Cyt b* gene. The evolutionary history of the listed *Tor* species was inferred using the maximum likelihood method and Tamura–Nei model [126]. These analyses were conducted using MEGA X (Version 10.1) [114,115]. Percentages on the branches represent bootstrap values.

**Figure 6 biology-10-00286-f006:**
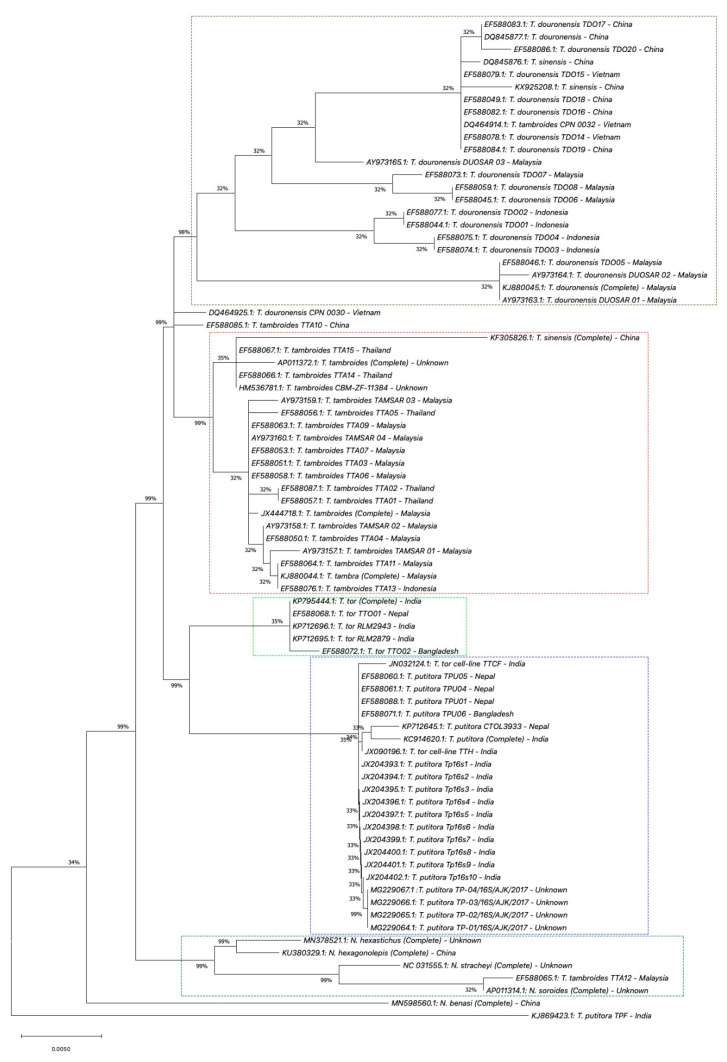
Phylogenetic tree of *Tor* species using the *16S rRNA* gene. The evolutionary history of the listed *Tor* species was inferred by using the maximum likelihood method and Tamura–Nei model [126]. These analyses were conducted using MEGA X (Version 10.1) [114,115]. Percentages on the branches represent bootstrap values.

**Table 1 biology-10-00286-t001:** List of reported *Tor* species worldwide and their validity from two different sources: Eschmeyer’s Catalog of Fishes and Pinder et al. (2019).

No.	Species	Species Status According to Eschmeyer’s Catalog of Fishes as of 1 March 2021	Species Status According to Pinder et al. (2019)
1.	*Tor ater* (Robert, 1999)	Valid	Valid
2.	*Tor barakae* (Arunkumar and Basudha, 2003)	Valid	Valid
3.	*Tor chelynoides* (McClelland, 1839)	Not Valid Valid as *Naziritor chelynoides* (McClelland, 1839)	Not Valid
4.	*Tor dongnaiensis* (Hoang et al., 2015)	Valid	Valid
5.	*Tor douronensis* (Valenciennes, 1842)	Not Valid. Valid as *T. tambra* (Valenciennes, 1842)	Not Valid
6.	*Tor hemispinus* (Chen and Chu, 1985)	Not Valid Valid as *Neolissochilus hemispinus* (Chen and Chu, 1985)	Not Valid
7.	*Tor khudree* (Sykes, 1839)	Valid	Valid
8.	*Tor kulkarnii* (Menon, 1992)	Valid	Valid
9.	*Tor laterivittatus* (Zhou and Cui, 1996)	Valid	Valid
10.	*Tor macrolepis* (Heckel, 1838)	Not Valid Valid as *T. putitora* (Hamilton, 1822)	Not Valid
11.	*Tor malabaricus* (Jerdon, 1849)	Valid	Valid
12.	*Tor mekongensis* (Hoang et al., 2015)¶	Valid	Not Valid Valid as *T. tambra* (Valenciennes, 1842)
13.	*Tor mosal* (Hamilton, 1822)	Valid	Valid
14.	*Tor polylepis* (Zhou and Cui, 1996)	Valid	Valid
15.	*Tor putitora* (Hamilton, 1822)	Valid	Valid
16.	*Tor qiaojiensis* (Wu, 1977)	Not Valid Valid as *Neolissochilus qiaojiensis* (Wu, 1977)	Not Valid
17.	*Tor remadevii* (Madhusoodana Kurup and Radhakrishnan, 2011)	Valid	Valid
18.	*Tor sinensis* (Wu, 1977)	Valid	Valid
19.	*Tor soro* (Valenciennes, 1842)	Not Valid Valid as *Neolissochilus soro* (Valenciennes, 1842)	Not Valid
20.	*Tor tambra* (Valenciennes, 1842)	Valid	Valid
21.	*Tor tambroides* (Bleeker, 1854)	Valid	Valid
22.	*Tor tor* (Hamilton, 1822)	Valid	Valid
23.	*Tor yingjiangensis* (Chen and Yang, 2004)	Valid	Valid
24.	*Tor yunnanensis* (Wang, Zhuang, and Gao, 1982)	Not Valid Valid as *Folifer yunnanensis* (Wang, Zhuang, and Gao, 1982)	Not Valid

Note: ¶ symbol is represent uncertain species.

**Table 2 biology-10-00286-t002:** List of reported *Tor* species in Southeast (SE) Asian countries.

No.	Species	Common Name(s)	Reported Geographical Distribution	Status (Based on Eschmeyer’s Catalog of Fishes as of 1 March 2021 and Pinder et al. (2019))	IUCN Red List of Threatened Species^TM^ (Version 2020-3) [11]		References
					Species Status	Population Status	
1.	*T. ater* (Robert, 1999)		Laos	Valid	Near Threatened	Unknown	[36]
2.	*T. dongnaiensis* (Hoang et al., 2015)	Dongnai mahseer	Vietnam	Valid	Near Threatened	Unknown	[15]
3.	*T. douronensis¶* (Valenciennes, 1842)	Semah (Indonesia/Borneo Island), Pelian (Malaysia)	Malaysia, Indonesia	Not Valid/Valid as *T. tambra*/Unknown/Questionable/Data Deficient	Not Listed	Not Listed	[1,38]
4.	*T. laterivittatus* (Zhou and Cui, 1996)		Laos, China	Valid	Data Deficient	Decreasing	[16,37]
5.	*T. mekongensis¶* (Hoang et al., 2015)		Vietnam	Not Valid/Valid as *T. tambra*/Unknown/Questionable/Data Deficient	Not Listed	Not Listed	[15]
6.	*T. mosal* (Hamilton, 1822)	Mosal mahseer, Copper mahseer	Myanmar, India	Valid	Data Deficient	Unknown	[3,34]
7.	*T. putitora* (Hamilton, 1822)	Himalayan mahseer, Golden mahseer, Putitora mahseer	Myanmar, Afghanistan, Bangladesh, Bhutan, India, Nepal, Pakistan	Valid	Endangered	Decreasing	[17]
8.	*T. sinensis* (Wu, 1977)	Pba daeng (Laos), Red mahseer	Vietnam, Laos, China	Valid	Vulnerable	Unknown	[15,39]
9.	*T. soro* (Valenciennes, 1842)	Kancra or Gemo fish	Indonesia	Not Valid/Valid as Other Species/Unknown/Questionable/Data Deficient	Not Listed	Not Listed	[33,38]
10.	*T. tambra* (Valenciennes, 1842)	Keureling (Indonesia), Pba tohn (Laos)	Malaysia, Thailand, Indonesia	Valid	Data Deficient	Decreasing	[3,35,38]
11.	*T. tambroides* (Bleeker, 1854)	Kelah, Empurau (Malaysia) Jurung (Indonesia)	Malaysia, Thailand, Indonesia	Valid	Data Deficient	Unknown	[3,35,40]
12.	*T. tor* (Hamilton, 1822)		Myanmar, Pakistan, Nepal, Bhutan, India, Bangladesh	Valid	Data Deficient	Unknown	[17,34]
13.	*T. yingjiangensis* (Chen and Yang, 2004)		Myanmar, China	Valid	Data Deficient	Unknown	[26,28]

Note: ¶ symbol is represent uncertain species.

**Table 3 biology-10-00286-t003:** The morphological features of SE Asia’s *Tor* species. The figures in this table were drawn based on an actual sample of fish, descriptions recorded, or redrawn from photos: *T. ater* [36]; *T. dongnaiensis* [15]; *T. douronensis* [87]; *T. laterivittatus* [88]; *T. mekongensis* [15]; *T. mosal* [89]; *T. putitora* [89]; *T. sinensis* [15]; *T. tambra* [13]; *T. tambroides,* our picture and [90]; *T. tor* [89]; *T. yingjiangensis* [28]. Note: ¶ symbol is represent uncertain species.

Species	Reported Maximum Size (Standard Length for Adult, cm)	Mouth Position	Lower Median Lobe(Scale Bars Represent the Length of Lower Median Lobe)	Body Color	Lateral Scales	Distinctive Features	Reference
*T. ater*(Robert, 1999)	~30	Sub-terminal 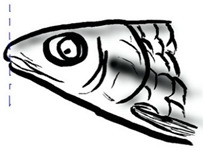	Short 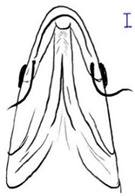	Dark brown with the presence of a dark longitudinal stripe	30–31	The smallest in body size among SE Asia’s *Tor* species. Short median lobe and longitudinal dark stripe on the body. 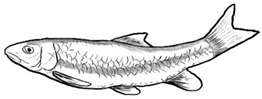	[52]
*T. dongnaiensis*(Hoang et al., 2015)	~40	Sub-terminal 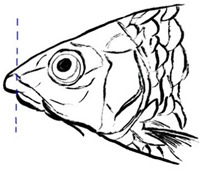	Long 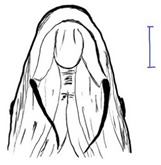	Silver gray and yellowish in sub-adult	23–24	Long lower median lobe without upper median projection. 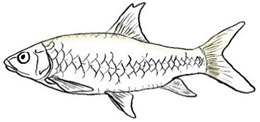	[15]
*T. douronensis¶* (Valenciennes, 1842)	~100	Sub-terminal 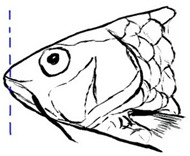	Medium 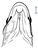	Silvery, darkish above and dark fins	24–27	Short lower median lobe and silvery and yellowish in color. 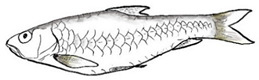	[2]
*T. laterivittatus*(Zhou and Cui, 1996)	~60	Sub-terminal 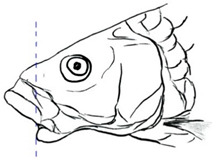	Long 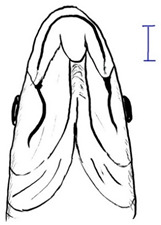	Dark green to silver with the presence of longitudinal stripe	25–27	Long lower median lobe without upper median projection and longitudinal dark brown stripe along the middle side of the adult body. 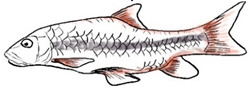	[91]
*T. mekongensis¶*(Hoang et al., 2015)	~33	Sub-terminal 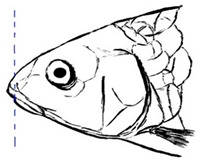	Medium 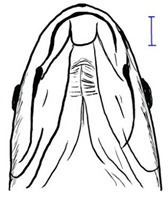	Silver gray	23	Short lower median lobe, blunt rostral hood, and silvery in color. 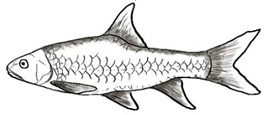	[15]
*T. mosal*(Hamilton, 1822)	~270	Terminal 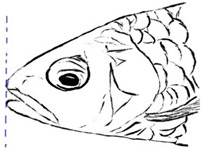	Long 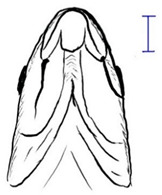	Delicate yellowish shade below, caudal reddish orange	23–26	Terminal mouth position with long lower median lobe, yellowish color on the lower body, and fin ray counts: 13 dorsal fin rays, 17 pectoral fin rays, and 8 anal fin rays. 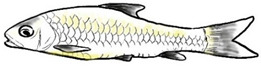	[3]
*T. putitora*(Hamilton, 1822)	~270	Sub-terminal 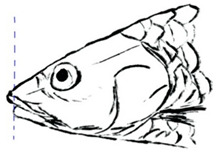	Varying in length 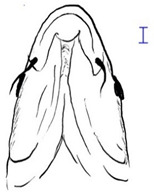	Reddish sap green, light orange fading to silvery white	23–28	Blunt rostral hood and dark longitudinal stripe in the middle of the side of the body. The color of the caudal, pelvic, and anal fins is reddish gold. 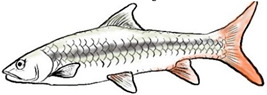	[92]
*T. sinensis*(Wu, 1977)	~46	Sub-terminal 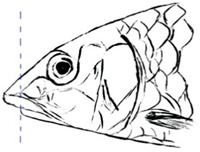	Long 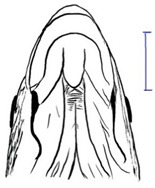	Darkish on the back and brownish or bronzy beneath, a dark longitudinal stripe	23–28	Long lower median lobe without upper median projection, pointed rostral hood, and a slate gray longitudinal stripe along the side of the body from head to caudal base. 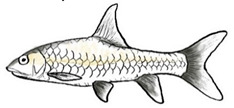	[37]
*T. tambra* (Valenciennes, 1842)	~100	Sub-terminal 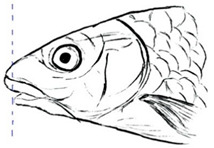	Short 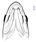	Olive or dark olive, reddish	22–24	Short lower median lobe, blunt rostral hood, and reddish body color. 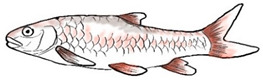	[13]
*T. tambroides*(Bleeker, 1854)	~100	Sub-terminal 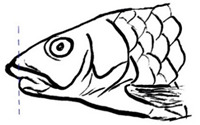	Long 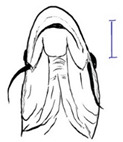	Silver bronze and reddish with dark fin	23–26	Long lower median lobe, present with upper median projection, pointed rostral hood, and reddish body color. 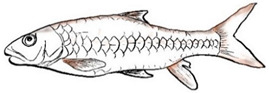	[93]
*T. tor*(Hamilton, 1822)	~200	Terminal 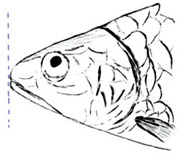	Varying in length 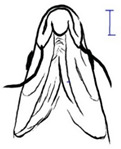	Grayish green	22–28	Small head and mouth, blunt rostral hood, reddish fin, silvery abdomen, and dark gray dorsal side. 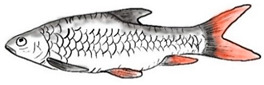	[12]
*T. yingjiangensis*(Chen and Yang, 2004)	~20	Terminal 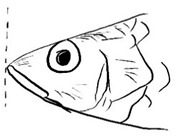	Short 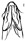	Yellowish	24–26	Longer head, conical shape of the head, and a slightly convex body. All fins are yellowish, and there is no mid-lateral line. 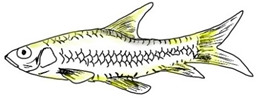	[28]

**Table 4 biology-10-00286-t004:** Suggested valid *Tor* species in SE Asian region.

No.	Species Name
1.	*T. ater* (Robert, 1999)
2.	*T. douronensis* (Valenciennes, 1842)
3.	*T. laterivittatus* (Zhou and Cui, 1996)
4.	*T. mosal* (Hamilton, 1822)
5.	*T. putitora* (Hamilton, 1822)
6.	*T. sinensis* (Wu, 1977)
7.	*T. tambra* (Valenciennes, 1842)
8.	*T. tambroides* (Bleeker, 1854)
9.	*T. tor* (Hamilton, 1822)
10.	*T. yingjiangensis* (Chen and Yang, 2004)

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
