# Peer review of "A Current Update on the Distribution, Morphological Features, and Genetic Identity of the Southeast Asian Mahseers, Tor Species"

_biology, 2021, doi:10.3390/biology10040286_

Round 1

Reviewer 1 Report

The authors have done well addressing my concerns raised in my initial review of this paper. 

Author Response

Reviewer 1 Comment

The authors have done well addressing my concerns raised in my initial review of this paper. 

Authors Response

Thank you very much for the comment given previously, which help to improve the quality of this manuscript.

Reviewer 2 Report

I generally agree with this revise. Please correct the following points.

Fig.3-6:
Please correct the taxon label so that it is easy to see.
For example, display only the species name and accession number.

Please put an outgroups.

Please make it a non-linearized phylogenetic tree.

Please insert the scale bar.

Reviewer 3 Report

This is a comprehensive review but there is a lack of structure and at points there is extensive irrelevant text. The manuscript is very long and I wonder if it would be better suited to more that one manuscript that did each section better justice. I was surprised to see new phylogenetic analysis but no methods section. I would like to see better justification for use of sources for the databases. Figures and tables need clearer descriptions and the phylogenetic trees would be more useful if they could be simplified and made more legible. I'm not sure on the need for some of the tables, e.g. Table 4. The distribution section seems to be more of a discussion of knowledge on each country rather than distributions. The country sections read well but I expected from the title for these to be structured around the Tor species rather than by country. The manuscript is currently lacking any conclusions and instead ends with a section on Application of biotechnology, which does not seem relevant to this manuscript to me and is a poor place to be adding new information.

Standard formatting needs to be taken care of for example several mention of 'and coworkers' or 'and colleagues' should be et al. for simplification. Also after the first mention of a species name it should always have the genus abbreviated e.g. T. tambra, unless the genus is the first word of a sentence.  

I have provided line by line comments for suggestions to improve the manuscript and for some corrections.

397 first 'at' should be 'et'

Round 2

Reviewer 3 Report

Whilst the manuscript is improved there is still a large amount of unnecessary and repeated text. I have provided extensive feedback and also some English editing as some of the statements intentions were misleading due to a lack of clarity in the language, I have not been able to amend all errors.

24 ‘the’ not needed before SE

26 ‘characteristic’ should be characteristics as there are multiple listed

26 ‘lateral scales number’ should be reordered to ‘number of lateral scales’, for grammatical sense

30 ’its’ should be ‘their’, as refers to multiple species

34 If mahseers is only these three genera then remove the word ‘including’ which suggests there are others that are not listed here

51 If you are referring to the word ‘scale’ as a measure of size, it should be removed as this doesn’t make grammatical sense

52 No hyphen needed in ‘freshwater’

53 ‘this three’ should be ‘these three’

54 Either ‘Tor species are an attractive sport fish’ or ‘Tor species are attractive for sport fishing’ but the current sentence is not grammatically correct

54 change ‘high’ to ‘highly’

55 ‘increase’ should read ‘increasing’

64 insert the word ‘in’ between upstream and rivers

80 phrasing ‘In recent ten years’ does not make sense, ‘In the last ten years’ would work

80 I assume the authors mean ‘DNA sequencing’ here

113 – 115 Remove this fist sentence which is not related to Tor. Could include this type of information in the following sentences where relevant i.e. “Peninsular Malaysia is home to two Tor species, T. tambroides and T. tambra (see Table 2), where it is found in x out of 89 major river basins.”. The information for ’22 major river basins in Sawawak’ is already included on line 169

131 This is not a logical conclusion that ‘Due to deforestation, the Tor species has been reported only in…’, explain what is meant here.

135 – 136 Tell the reader since when has is not been caught?

137 inhibit? Do you mean inhabit?

157 ‘is’ should be are

174 remove ‘taken’

229 it’s not clear what ‘3S’ is here

238 – 242 These first two sentences are not relevant to the topic, remove.

246 – 247 The figure is T, tambroides, so this citation should come directly after that, not here at the end of the sentence after T. tor.

249 ‘big issue’ is a very subjective term. Also a lack of studies does not equate to species disappearing or rapidly decreasing, qualify or edit this sentence.

253 no hyphen is needed in ‘freshwater’

254 – 256 List of river names not useful information for the reader to learn about Tor, remove

262 – 263 this is a repeat of the info on line 253, remove from one location

266 Remove unnecessary text ‘Cambodia comprises inland and marine water systems.’

266 – 273 These sentences are not related to Tor and can all be removed. Stick to the topic of the review, the objective of this paper is not about the geography of South East Asian rivers. Only discuss in relation to Tor.

274 ‘almost unavailable’ is a poor phrase, they are either available or unavailable, which is it?

279 – 282 Again, irrelevant information, first two sentences can be removed.

283 it doesn’t seem surprising to me that Indonesia would have similar species to Malaysia when they share a border. Re phrase.

315 – 316 Again this first sentence is not relevant, instead could incorporate into second sentence e.g. A recent study reported that T. tambroides and T. tambra inhabit Brunei’s four major river systems [84]; Belait River, Tutong 315 River, Temburong River, and Brunei River.’ If that is true

318 What is recent? This is a very subjective term. What year was the last study?

318 ‘the Tor species’ which one? Or do you mean Tor species generally in which case delete ‘the’

319 remove word ‘species’ after ‘extinct’

321 – 322 Saying not connected to mainland, hence, no Tor species contradicts your statement from 282 – 283 states ‘Although Indonesia is separated from mainland SE Asia, Indonesia somehow shares a similar Tor species with some SE Asian countries’.

325 ‘the’ cannot be used before Tor here because that would refer to a single Tor species, whereas you are talking about determining between, therefore multiple Tor species, remove word ‘the’

326 ‘biology researchers’ is very vague, what type of biologists – taxonomists?

328 – 329 ‘difficulty to validate’ has led to ‘inconsistent validation’ seems a bit of a circular point.

330 remove ‘and’ between ‘identification’ and ‘measurement’, replace with a comma

334 ‘T. ater is the smallest Tor species as compared to other SE Asia’s Tor species.’ Could be simplified to ‘T. ater is the smallest of the SE Asian Tor species.’

4.1 This whole section provides no text that is not already in the table and is just a repeat of information.

343 Hoang et al 2017 is not in the reference list.

343 remove the word ‘study’ or ‘analysis’ both are not necessary

Table. 3 there appear to be blue scale bars but no indications as to what the relevant scale is.

Table 3. remove ‘Have’ from the beginning of descriptive features test

350 – 352 – ‘Nonetheless, a recent review by Pinder et al. (2019), suggests that this species remains a valid species, and thus a large-scale study is required to validate this species' uncertainty  [30].’ That doesn’t seem to be what Pinder et al say about T. dongnaisensis ‘T.mekongensis is currently considered to be a questionable synonym of the wide-ranging T. tambra (see Walton et al. 2017).’ They also say it’s potentially the same, also it doesn’t really make sense to just cite another review instead of citing the actual source of information i.e. Walton et al 2017

434 T. yingjiangensis is said to be least studied but 380 – 381 You’ve already said T. laterivittatus is the least studied. They can’t both be. Rephrase one of these.

573 and colleagues should be et al,.

621 A conclusion of this review of distribution, morphology and phylogeny is not ‘and are important as food fish across SE Asia’ remove. This information is already stated in your introduction, this is not a relevant place for that.

621 – 623 ‘The king  of rivers or mahseers comprises three genera, including Tor, Neolissochilus, and Naziritor, under the  Cyprinid family.’ This is also introductory information and is not a conclusion of this review, remove.

623 ‘There are 12 Tor species’ sounds very definitive for a conclusion following a review showing how complex and understudied they are and that you just presented a table suggesting only 9

624 – 625 ‘This review outlines the distribution, morphology, and genetic identification of the SE Asian’s Tor species,’ Again this sounds like introductory information and the conclusion is not the right place for this.

I’m surprised that the conclusion doesn’t touch upon the fact that the two least studied Tor – laterivittatus and yingjiangensis have unknown distribution and they are not included in the phylogenetic analysis presumably due to a lack of samples. This is the kind of specific call to action I would expect to see.

Author Response

Dear Reviewer,

Thank you very much

This manuscript is a resubmission of an earlier submission. The following is a list of the peer review reports and author responses from that submission.

Round 1

Reviewer 1 Report

Only one year ago, an international team of authors published the first comprehensive review of the genus Tor in Reviews in Fish Biology and Fisheries (Pinder et al., 2019). This review synthesised all available literature and recent novel research by the research team, to present a revised list of valid Tor species and the current state of knowledge for each species, including their known distributions and population status. For each species, the outstanding gaps in knowledge were also identified, and their population threats and conservation prospects outlined.
Specifically, this review provided the basis for researchers to challenge and enhance the knowledge base necessary to conserve these freshwater icons in an era of unprecedented environmental changes. 

This current MS does not present any new insight to resolve the outstanding uncertainties presented by Pinder et al. It does not challenge any of the conclusions drawn by the 2019 review and unfortunately contributes no further value to the knowledge base. 

The genus Tor remains highly threatened and it is imperative that future reviews build on the current state of knowledge. The publication of the current MS would only add further confusion to the evidence base and risk jeopardising the effective conservation of these iconic species. 

Reviewer 2 Report

This is an interesting study on an important group of fish, with difficult taxonomic classification, therefore this study contributes to the better understanding. To the best of my knowledge, the review is comprehensive and quite exhaustive.

There’s issues with the use of the English writing throughout the manuscript. I have highlighted some below, but this is not exhaustive. I suggest that the use of English is revised by getting a native English speaker to revise.

Line 22: “as a highly esteemed..”
Line 25: cultivation instead of “culture”

Both the summary and the abstract summarise well what we know on this taxa of ish. What it needs here towards the end of the abstract , is to highlight what the main issues are in relation to the study if this group; major gaps in understanding etc. The last statement (lines 45-47) does not tell us much..why would this review offer a re-kindling of research interest on this taxa?

Line 67: “tons”

Ln 74: why important in ecosystems?
Ln 88: “caught from the wild”
ln: 166 at an alarming rate
186: where is the data from used to generate map?
187: locations
189-192: not clear why the population is limited – rephrase
197: among Thai people
196-202: References?
Note: throughout the paper, the full species name should be mentioned on first mention ( eg Tor dongnaiensis) and from then on it should be T. dongnaiensis. Same with all species mentioned.
269-270: remove or rephrase – most countries contribute to some extent to fish production

A GIS map showing distribution of the genera would be useful as it would allow some biogeographic conclusions to be drawn.

Table 3 is very useful and an excellent summary of the main morphological features.

394: why did they postulate that it is a valid distinct species?

The review, overall, is very good. The only aspect that is significantly lacking is the final part (lines 562 onwards). It is rather weak and needs more convincing as to why this taxa merits further focus. You state at the start that there's a lack of publications, but you go on and list a long list of studies in this taxa. There are other taxa out there that have received even less attention. Perhaps be a bit more forceful here.
Plus, you need more on the general outlook of this taxa and , very importantly, identified the major gaps and show some insights in terms of what the information you gathered show.

Reviewer 3 Report

I think this review paper is important to conservation of the mahseer, Tor species.

I want you check the following.

Table 1 No.12  Do you forget writing "not valid"?

Table 3 Is it possible that you prepare the photos of Tor species?

Table 3 The description of Mouth position & Lower Median lobe is  equivocal. Is it possible that you state numerically?

4 Genetic identification

Is it possible that you prepare the phylogeny of Tor species and the results of SSR analyses (eg. STRCTURE)?